# Dietary choline in gonadectomized kittens improved food intake and body composition but not satiety, serum lipids, or energy expenditure

Hannah Godfrey[1], Alexandra Rankovic[2], Caitlin E. Grant[1], Anna Kate Shoveller[3], Marica Bakovic[4], Sarah K. Abood[1], Adronie Verbrugghe[1]*

1 Department of Clinical Studies, Ontario Veterinary College, University of Guelph, Guelph, Ontario, Canada, 2 Department of Biomedical Sciences, Ontario Veterinary College, University of Guelph, Guelph, Ontario, Canada, 3 Department of Animal Biosciences, Ontario Agricultural College, University of Guelph, Guelph, Ontario, Canada, 4 Department of Human Health and Nutritional Science, College of Biological Science, University of Guelph, Guelph, Ontario, Canada

☯ These authors contributed equally to this work.
¤ Current address: Sit, Stay, Speak Nutrition LLC., Dimondale, Michigan, United States of America
* averbrug@uoguelph.ca

**Data Availability Statement:** All data files are available from the Scholars Portal Dataverse for the

## Abstract

Gonadectomy is a major risk factor for feline obesity. The lipotropic effects of choline have demonstrated benefits for growth and carcass composition in livestock. The consumption of supplemental choline on body weight (BW), body composition, lipid metabolism, energy expenditure (EE), and serum satiety hormones were evaluated in 15 gonadectomized male kittens. Kittens were offered a base diet formulated for growth (3310mg choline/kg dry matter [DM]) to daily energy requirements (DER) over an 11-week acclimation. Post-gonadectomy, kittens were assigned to a base diet (CONTROL, n = 7) or choline group (base diet with additional choline at 300mg/kg $BW^{0.75}$ as a top dress) (CHOLINE, n = 8). For 12-weeks post-neuter, kittens were offered three times their DER over three meals to mimic *ad libitum* feeding. At week -1 and 12, body composition was assessed using dual energy x-ray absorptiometry (DXA), 24-hour indirect calorimetry was performed for EE and respiratory quotients (RQ), and fasted serum samples were analyzed for lipid compounds and satiety hormones. Daily food intake (FI) and weekly BW were measured. Data was analyzed as a repeated measures of variance (ANCOVA) using the GLIMMIX procedure with time and group as fixed effects. CHOLINE had lower mean daily FI and lower rates of BW accretion (P<0.05) in contrast to CONTROL. All absolute body composition data increased over time for both groups, with lower increases in total tissue mass (P = 0.031) and fat mass (P = 0.005) in CHOLINE. Serum satiety hormones and lipid compounds did not differ (P>0.05) between groups, but both groups experienced a decrease in low-density lipoproteins and increase in high-density lipoproteins (P<0.05). Primary substrate utilization showed lipid use when fasted and use of protein or mixed macronutrients in the fed state. Fed state EE decreased post-gonadectomy (P = 0.004), however, CHOLINE did not affect total EE or RQ. These results suggest that supplemental dietary choline reduces FI, BW, and fat mass

University of Guelph (doi.org/10.5683/SP3/OJZEVR).

**Funding:** This study was supported by the Natural Sciences and Engineering Research Council (NSERC) Collaborative Research and Development [https://www.nserc-crsng.gc.ca/professors-professeurs/rpp-pp/crd-rdc_eng.asp] in partnership with Elmira Pet Products (Elmira, ON, Canada) [http://www.elmirapet.com/] through a grant awarded AV (CRDPJ 472710-16). The Choline Chloride supplement was kindly provided by Balchem (New Hampton, NY, USA). The funders had no role in study design, data collection and analysis, decision to publish, or preparation of the manuscript.

**Competing interests:** The authors declare no conflicts of interest. AV is the Royal Canin Veterinary Diets Endowed Chair in Canine and Feline Clinical Nutrition and declares that they serve on the Health and Nutrition Advisory Board for Vetdiet. AV has received honoraria and research funding from various pet food manufacturers and ingredient suppliers. HG declares that they have paid engagements with pet food companies within Canada. AR declares that they have participated in paid internships and engagements with pet food companies within Canada. CG holds the Nestlé Purina Professorship in Companion Animal Nutrition at the Ontario Veterinary College, is the owner of Grant Veterinary Nutrition Services and consults with Simmons Pet Food. The Ontario Veterinary College received funding from Nestlé Purina Proplan Veterinary Diets to support a Registered Veterinary Technician in Clinical Nutrition, who assisted with this study. AKS was previously employed by P&G and Mars Pet Care, serves on the Scientific Advisory Board for Trouw Nutrition, and has received honoraria and research funding from various commodity groups, pet food manufacturers, and ingredient suppliers. SKA is the owner of Sit, Stay Speak Nutrition LLC and provides nutrition consultation to industry partners. This does not alter our adherence to PLOS ONE policies on sharing data and materials.

and may help to reduce the propensity of weight gain and subsequent obesity in gonadectomized feline populations.

## Introduction

Obesity, a multi-factorial disease, is estimated to impact 11 to 61% of domestic cats globally [1–5]. Positive energy balance can be a result of many factors, including food intake (FI) being greater than total energy requirements and low total daily energy expenditure (EE). Together, these cause fat deposition and weight gain [6, 7]. Many risk factors for obesity have been identified in cats, and gonadectomy is consistently reported as one of the significant risk factors [8–11]. Over 79% of domestic cats are gonadectomized, according to a study from Perrin [12], and this trend will continue due to the many benefits of spay/neuters such as disease prevention, mitigating unwanted behaviours, and reducing unwanted pregnancies [13–15]. Overall, obesity prevention strategies may need to be targeted at this vulnerable population.

Choline's role in lipid metabolism makes it a nutrient of interest for obesity prevention and treatment. Deficiency studies in both kittens and adult cats concluded that choline is an essential nutrient for cats [16–18]. Currently, the recommended intake for growing cats is 133 mg choline/kg $BW^{0.67}$ [19] and this recommendation was based on the three choline deficiency studies in growing kittens and adult cats [16–18]. Choline plays a role as phosphatidylcholine (PC), which is required for the assembly of very-low-density lipoproteins (VLDL) for the endogenous transport of cholesterol (CHOL) and triglycerides (TAG) from the liver [20–23]. Triglycerides and CHOL esters form the core of lipoproteins, held together by phospholipids, most commonly PC. Due to this role of choline, deficiency is a known factor in developing non-alcoholic fatty liver in many species, including cats [16]. In obese cats fed to maintain body weight, choline supplementation at 5.1 times the NRC [19] recommended allowance resulted in increased blood CHOL, TAG, and VLDL which could indicate increased mobilization from the liver [24].

As a methyl donor, the choline oxidation product, betaine, reduces plasma homocysteine levels by the re-methylation of homocysteine to methionine [21]. Methionine is then converted to the universal methyl donor, S-adenosylmethionine (SAM), which methylates DNA, RNA, proteins, and lipids [25]. Decarboxylated SAM generates spermine from spermidine which is required for growth [25]. Through this one-carbon metabolism cycle, choline increases creatine synthesis, which is used in the delivery of energy to tissue [26].

Supplementation of betaine had beneficial effects on body composition, i.e., lower fat mass and greater lean mass, with lower FI than control fed finishing pigs, ducks, and broilers [27–29]. Similarly, choline supplementation decreased BW gain in rats [30] and resulted in lower body fat and higher body protein content in guinea pigs [31]. Choline, as acetylcholine, may have implications with satiety hormones through an inhibitory effect as seen with the anorexigenic hormone ghrelin [32]. However, the mechanisms behind those changes are less clear and the effects of choline on EE and energy partitioning in cats have not been investigated.

This study aimed to investigate the effects of additional choline supplementation on parameters impacted by gonadectomy and obesity such as FI, BW, body composition, fasted serum lipid and lipoprotein profile, fasted serum satiety hormones, EE and respiratory quotient (RQ). It is hypothesized that cats fed CHOLINE will experience lower FI, BW, BCS, and body fat mass, yet higher lean mass than cats fed CONTROL. Although choline mobilizes VLDL and TAG out of the liver in the fasted state, choline also improves lipoprotein uptake into the

muscle [33]. Therefore, it is hypothesized that these variables will be lower with choline supplementation than in the control group. It is hypothesized that RQ in choline supplemented cats will indicate enhanced fatty acid oxidation and that EE will be increased in the choline supplemented group. Finally, choline supplementation may alter satiety hormones by reducing fasted serum leptin and ghrelin levels and increasing peptide YY (PYY), Glucagon-like peptide-1 (GLP-1), and gastric inhibitory protein (GIP) for improved regulation of appetite, FI, and EE.

## Materials and methods

### Animals

All experimental procedures were approved by the University of Guelph Animal Care Committee (AUP #4118) and were in accordance with national and institutional guidelines for the care and use of animals in research. Sixteen intact male, domestic shorthair, 12- to 13-week-old kittens (Marshall's Bio Resources, Waverly, NY, USA) were housed at the Animal Biosciences Cattery at the Ontario Agricultural College of the University of Guelph (Guelph, ON, Canada). At week -1, cats had an initial mean (±SEM) BW of 3.85 ± 0.067 kg. Body condition scores (BCS) were assessed between 4 and 6 on a 9-point scale [34]. Prior to the study, all cats were deemed healthy based on medical history, physical exam, as well as serum biochemistry profile and complete blood count.

### Housing

All cats were housed together in an indoor, free-living environment (7.1 m x 5.8 m). Enrichment was provided through scratching posts, cat trees of differing size and height, hiding boxes, perches, beds, and a variety of toys. Cats received human interaction in the form of voluntary brushing, petting, and playing for a maximum of two hours per day for five days by familiar individuals. Cats had access to water *ad libitum*. Temperature and humidity were maintained at an average of 22.8°C and 48.4% respectively. A 12-hour light and dark cycle with lights turning on at 0700 h and turning off at 1900 h was maintained at all times. The room was cleaned and sanitized daily, and litter boxes scooped, and litter added, if needed, twice daily.

### Diet and choline supplementation

Cats were fed a commercially available extruded diet, Nutram Sound Balanced Wellness® Kitten Food Chicken Meal and Salmon Meal Recipe (Elmira Pet Products, Elmira, ON, Canada), formulated for growth according to The Association of American Feed Control Officials (AAFCO) [35] as base diet throughout the study. Nutrient analyses are summarized in Table 1. Proximate analysis and vitamin analysis were performed by Bureau Veritas (Mississauga, ON, Canada) using appropriate AOAC (Association of Official Analytical Chemists) and AOCS (American Oil Chemist Society) methods [36]. Moisture and dry matter were determined by gravimetry (AOAC 930.15). Crude protein was quantified using the Kjeldahl method (N x 6.25) (AOAC 992.15), crude fat by gas chromatography/flame ionization detector (AOAC 996.06), and ash and crude fibre by gravimetry (AOAC 923.03 and AOCS Ba 6a-05, respectively). Metabolizable energy was calculated using the Modified Atwater equation and nitrogen free extract by subtracting crude protein, crude fat, crude fibre, moisture, and ash from 100 [19]. Choline was analyzed by the enzymatic method (AOAC 999.14), and vitamin B12, vitamin B6, and vitamin B9 were also analyzed using microbiological assays (AOAC 999, 986.23, and 2004.05, respectively) [36]. Dietary amino acid concentrations are summarized in

**Table 1. Dietary analysis (as fed) of the base diet, a commercial extruded kitten food, fed to all kittens in a choline supplementation study.**

|  | Nutrient | Content, as fed |
|---|---|---|
| **Proximate Analysis** | **Moisture, %** | 5.60 |
|  | **Dry Matter, %** | 94.40 |
|  | **Crude Protein, %** | 36.26 |
|  | **Fat, %** | 17.80 |
|  | **Crude Fibre, %** | 2.90 |
|  | **Ash, %** | 8.50 |
|  | **NFE[1], %** | 28.94 |
|  | **ME[2], kcal/kg** | 3795 |
| **Vitamins** | **Choline, %** | 0.312 |
|  | **Vitamin B1, %** | 0.0013 |
|  | **Vitamin B6, %** | 0.00113 |
|  | **Vitamin B9, %** | 0.0004 |
| **Amino Acids** | **Alanine, %** | 2.35 |
|  | **Arginine, %** | 2.50 |
|  | **Aspartic Acid, %** | 3.19 |
|  | **Glutamine, %** | 5.05 |
|  | **Glycine, %** | 3.52 |
|  | **Histidine, %** | 0.85 |
|  | **Isoleucine, %** | 1.40 |
|  | **Leucine, %** | 2.62 |
|  | **Lysine, %** | 2.16 |
|  | **Phenylalanine, %** | 1.62 |
|  | **Proline, %** | 2.59 |
|  | **Serine, %** | 1.54 |
|  | **Threonine, %** | 1.40 |
|  | **Tyrosine, %** | 1.20 |
|  | **Valine, %** | 1.64 |
|  | **Methionine, %** | 0.79 |
|  | **Cysteine, %** | 1.34 |
|  | **Tryptophan, %** | 0.12 |
|  | **Taurine, %** | 0.24 |

[1]Nitrogen Free Extract (NFE) = 100 –[crude protein + crude fat + crude fibre + moisture + ash] [19].

[2]Metabolizable Energy (ME) calculated using the Modified Atwater Equation: ME = 10 * [(3.5 * % Crude protein) + (8.5 * % Fat) + (3.5 * % NFE)] [19].

Ingredients: Chicken meal, salmon meal, deboned chicken, oatmeal, brown rice, chicken fat (preserved with mixed tocopherols), pearled barley, split peas, peas, *Medicago sativa* extract (active substance: Alfalfa polysaccharide), salmon oil, dry chicken hydrolysate, flaxseed, apples, carrots, pumpkin, dried whey protein concentrate, choline chloride, potassium chloride, pomegranate, cranberries, chicory root extract, dried kelp, taurine, DL-Methionine, vitamins & minerals (vitamin E supplement, niacinamide, vitamin A supplement, thiamine mononitrate, D-calcium pantothenate, pyridoxine hydrochloride, riboflavin, beta-carotene, vitamin D3, folic acid, D-biotin, vitamin B12 supplement, zinc proteinate, ferrous sulfate, zinc oxide, iron proteinate, copper, sulfate, copper proteinate, manganese proteinate, manganese oxide, calcium iodate, sodium selenite), sodium chloride, psyllium, *Yucca schidigera* extract, lamb meal, spinach, peppermint, turmeric, ginger, rosemary extract.

Table 1 and were quantified using ultra performance liquid chromatography as previously described [37].

During an 11-week acclimation, cats were offered an amount of food to meet their daily energy requirement (DER) for growth. The expected BW was estimated using a kitten growth chart [38] and DER was calculated using the following equation [19]:

$$DER = 100 * BW^{0.67} * 6.7 * [e^{-0.189p} - 0.66]$$

Where $p$ = $BW_a/BW_e$
Where $a$ = Actual BW and $e$ = Expected BW

During the supplementation period, all cats were fed to mimic *ad libitum* feeding by providing food three times daily for 20 minutes at 0800 h, 1200 h, and 1600 h. The food amount provided totaled three times their calculated DER. Food not consumed was weighed and subtracted from food offered to determine FI. Daily calorie, protein and amino acid intakes were calculated for each individual kitten, using the individual daily FI and the dietary ME, protein and amino acid content, respectively.

Choline supplementation for the choline group was provided as PetShure® 97% choline chloride dry dietary form (Balchem, New Hampton, NY, USA). The base diet group did not receive additional choline. A stock solution was prepared by mixing 97% choline chloride (choline equivalence 72.3%) in distilled water at a concentration of 500 mg choline chloride/ml distilled water. Individual daily choline doses were provided for each cat at 300 mg choline/kg $BW^{0.75}$ and the appropriate amount was pipetted. This dose is based on previous research of a dose response study of choline in rats [39], and in growing pigs [40], as well as a preliminary study in obese adult cats [24]. The supplementation group was provided a small portion of food (20 g) with added choline solution prior to the 0800 h feeding times while the base diet group was provided the base diet only. Choline intake (ChI) was calculated for each individual kitten using the individual daily FI and the choline content of the diet. The additional supplemented choline was also added to determine the ChI for the choline group.

## Experimental design

For 11-weeks prior to the gonadectomy and supplementation period, cats were successfully acclimated to being handled and restrained by the researchers, to the housing environment, and indirect calorimetry chambers following the protocol in Gooding et al. [41]. Success of acclimation was measured by the scale for cat stress scores established by Kessler and Turner [42]. After successful acclimation (week -1), cats were blocked by BW into two groups and then the groups were randomly assigned to the CONTROL [mean (±SEM) BW of 3.75 kg ± 0.088 kg] or CHOLINE [mean (±SEM) BW of 3.81 kg ± 0.088 kg] group. Cats were then split into four groups of four cats (mean (±SEM) BW of 3.79 kg (± 0.034 kg) for the four groups) for a staggered randomized complete block design. Each group consisted of two CHOLINE cats receiving the base diet food and choline supplementation and two CONTROL cats receiving the base diet but no supplementation.

Baseline data collection occurred during week -1, including: 24-h indirect calorimetry, BW, BCS, MCS, dual energy x-ray absorptiometry (DXA) and blood collection for serum biochemistry profiles, lipid profiles, and satiety hormones. After the baseline collections at the end of week -1, kittens were anesthetized for gonadectomy surgeries. Immediately following gonadectomy, modified *ad libitum* feeding of the base diet began for both groups and daily choline supplementation for the CHOLINE group for a period of 12 weeks. Daily FI and weekly BW, BCS, and MCS were taken. At the end of week 12 (endpoint), data collection was repeated, including: 24-h indirect calorimetry, BW, BCS, MCS, DXA and blood collections.

## Gonadectomy

Kittens were anesthetized and gonadectomized after the baseline collections. All kittens were sedated with dexmedetomidine hydrochloride (Dexdomitor, Zoetis, Kirkland, QC, Canada) (10 ug/kg) and Buprenorphine (Zoetis, Kirkland, QC, Canada) (20 ug/kg) intramuscularly. After sedation, an IV catheter was placed in the cephalic vein for intravenous propofol (Fresenius Kabi Canada Ltd, Richmond Hill, ON, Canada) (1–4 mg/kg) induction. Once induced, cats were intubated and maintained on isoflurane and oxygen. The surgical procedure consisted of a vertical incision made through one side of the scrotum then the tunic to extrude the testicle. Attachment of the tunic was avulsed from the head of the epididymis and the pedicle was auto-ligated with a figure-eight knot. The testicle was then incised from the pedicle and was repeated on the contralateral side. Scrotal incisions were left to heal by secondary intention. Following surgical procedures, cats were given Robenicoxib (Zoetis, Kirkland, QC, Canada) (2 mg/kg) subcutaneously. For three days following gonadectomy cats were administered Robenicoxib (Zoetis, Kirkland, QC, Canada) (6 mg per cat) orally once per day to control post-operative pain. Daily care following gonadectomy procedures for all cats remained unchanged.

## Body weight and body composition

Weekly BW, BCS, and MCS were measured after an overnight fast and prior to morning feeding. Body weights were measured using a Defender™ 3000 Xtreme W scale (OHAUS®, Parsippany, NJ, USA). Body condition scores and MCS were assessed by the same trained researcher (H.G.) throughout the study to reduce variability of measures [34]. The Gain:Feed and Gain: Choline ratios were calculated weekly for each kitten using the individual weight gain from each week and dividing by the individual weekly FI and weekly ChI of that respective week.

Dual energy x-ray absorptiometry scans were performed in the fasted state under sedation at week -1 and week 12. Sedation was achieved using Butorphanol (Zoetis, Kirkland, QC, Canada) (10 mg/ml) at a dose of 0.3 mg/kg BW [43], given intramuscularly in combination with intramuscular dexmedetomidine hydrochloride (Dexdomitor, Zoetis, Kirkland, QC, Canada) (0.5 mg/ml) at a dose of 0.001 mg/kg BW [43]. Triplicate scans were completed using a fanbeam DXA (Prodigy® Advance GE Healthcare, Madison, WI, USA) by a trained investigator (C.G.) on Small Animal Mode with the Thin Setting. Cats were positioned in dorsal recumbency with forelimbs extended cranially and repositioned as necessary between scans that were completed in approximately 10 minutes. Estimates for total tissue mass (TTM), fat mass, lean soft tissue mass (LSTM), bone mineral content (BMC), and bone mineral density (BMD) were obtained from the system software (enCORE Version 16; GE Healthcare, Madison, WI, USA). All three scans were averaged prior to statistical analysis. Sedation was reversed by atipamezole (Antisedan, Zoetis, Kirkland, QC, Canada) (5 mg/ml) at a dose 0.2 mg/kg BW [43].

## Blood collection and laboratory analyses

Whole blood was collected via jugular venipuncture using BD Vacutainer™ UltraTouch Push Button 23G x ¾ (Becton Dickson, Franklin Lakes, NJ, USA). Blood samples were taken in the fasted state under sedation immediately following the DXA scan at week -1 and week 12. Blood was collected into BD Vacutainer™ Venous Blood Collection Tubes: Serum Separating Tubes: Hemogard (Becton Dickson, Franklin Lakes, NJ, USA) to isolate serum. One of the tubes was used to collect 1.5 mL blood for satiety hormones and contained the DPP-IV inhibitor (Millipore Sigma, Billerica, MA, USA), protease inhibitor (Sigma-Aldrich, St. Louis, MO, USA), and Pefabloc SC inhibitor (Sigma-Aldrich, St. Louis, MO, USA) to prevent enzymatic degradation of satiety hormones before centrifugation and isolation of serum for GLP-1, GIP,

PYY, and ghrelin. Samples were immediately placed on iced and allowed to clot, prior to centrifugation at 2500 rpm for 15 minutes at 4°C (LegendRT, Kendro Laboratory Products 2002, Germany). Serum was pipetted and aliquoted into Fisherbrand™ Microcentrifuge Tubes (Thermo Fisher Scientific, Rochester, NY, USA) and stored at -20°C until further analyses.

Serum was sent to the Animal Health Laboratory, University of Guelph, for analysis of alanine aminotransferase (ALT), alkaline phosphatase (ALP), and gamma-glutamyl transferase (GGT), as well as for glucose, cholesterol, non-esterified fatty acids (NEFA), high-density lipoprotein cholesterol (HDL-C) and triacylglycerides (TAG). Samples were thawed, vortexed, and centrifuged prior to analysis on the Roche Cobas 6000 c501 analyzer (Roche Diagnostics, Basel, Switzerland). Low density lipoproteins (LDL) and VLDL were calculated as [VLDL = TAG/2.2] and [LDL = CHOL-HDL-VLDL] [44].

Satiety hormones were analyzed in serum by commercially available ELISA kits in triplicate. Ghrelin was assessed via the Ghrelin (Rat, Mouse) EIA Kit (Product # EK-031-31; Phoenix Pharmaceuticals Inc., Phoenix, AZ, USA) which was validated for use in cats previously [45]. Leptin was analyzed using a feline specific ELISA kit (Cat Leptin ELISA Kit, Product # MBS057075; MyBioSource LLC, San Diego, CA, USA). GLP-1, PYY, and GIP were assessed using kits validated for cats [46]: GLP-1 amide EIA Kit (Product # S-1359.0001; Peninsula Laboratories, Inc., San Carlos, CA, USA); Peptide YY EIA Kit (Product # S-1274.0001; Peninsula Laboratories, Inc., San Carlos, CA, USA); and Cat GIP ELISA Kit (Product # MBs064115-96; MyBioSource LLC, San Diego, CA, USA). All kits were run according to manufacturers' protocols. Quality control samples and standard curves were evaluated based on manufacturers' recommendations. The coefficients of variation (CV), or the ratio of the standard deviation to the mean, were assessed for each set of triplicates. If the CV was <20%, the results were deemed acceptable, and the average was used for further analysis. If the CV was >20% for triplicates, then duplicates were assessed with a CV <20% were used.

## Indirect calorimetry

Indirect calorimetry was performed in all cats at week -1 and week 12 to assess the effects of choline supplementation on EE and RQ. The original 146 cm x 60 cm x 89 cm plexi-glass indirect calorimetry chambers (Qubit Systems Inc., Kingston, ON, Canada) were reduced in size to 64 cm x 60 cm x 52 cm via wooden panel frames covered in polyethylene plastic wrap to remove dead space. Each chamber was equipped with a water bowl, litter box, resting box, blanket, and a cat toy; there was sufficient separation of feeding, sleeping, and elimination areas. Chambers were cleaned before and after all measurement periods.

Measurements were conducted in 24 h sessions following an overnight fast. Each session started with a 30-minute respiratory gas equilibrium period, 1.5 h fasted period, followed by fed and post-prandial periods following the three times feeding pattern (0800 h; 1200 h; and 1600 h). The chambers were an open circuit, ventilated system (Qubit C950 Multi Channel Gas Exchange, Qubit Systems Inc., Kingston, ON, Canada) with room air pulled through each chamber at flow rates set between 2.0–6.5 L/min to ensure $CO_2$ did not exceed 0.7%. Actual flow rate was measured using a mass flow meter (#F250, Qubit Systems Inc., Kingston, ON, Canada). Concentrations of $CO_2$ and $O_2$ gases within the chambers were measured via $CO_2$ (Q-S151, Quibit Systems Inc., Kingston, ON, Canada) and $O_2$ (Q-S102, Quibit Systems Inc., Kingston, ON, Canada) gas analyzers. Calibration of the calorimetry system, gas analyzers and mass flow meters, was performed via standard gas mixtures, nitrogen at 99.98% concentration and carbon dioxide at 1.01% (Praxair, Guelph, ON, Canada) at the beginning of each session and recalibrated if a drift of more than 1% was observed. Drierite (WA Hammond Drierite, Xenia, OH, USA) columns were placed immediately prior to the flow meters and the gas analyzers to ensure

only dry air was analyzed. The Quibit system software calculates RQ based on the ratio of $CO_2$ and $O_2$. Energy expenditure was calculated using the following equation [47]:

$$EE\ (kcal) = 3.94 \times O_2\ consumed\ (L) + 1.11 \times CO_2\ produced\ (L)$$

To adjust EE and RQ for food intake and body weight respectively, these variables were added as a covariate in the statistical analysis.

## Statistical analyses

Statistical analyses were performed using Statistical Analysis System (SAS Studio, 3.8, SAS Institute, Cary, North Carolina, United States). Results were analysed using the proc GLIM-MIX procedure as a repeated measures of variance (ANCOVA) model, with time and group as a fixed effect, cat as subject, and baselines were used as a covariate when significantly different. Data was assessed for normality using the Shapiro-Wilk test and log transformation was used as necessary to meet the assumptions of ANCOVA. A Tukey post-hoc adjustment using the covariance structure that resulted in the smallest Akaike information criterion value was used to separate means when the fixed effect was significant.

Area under the curve (AUC) for RQ data was calculated using the trapezoidal method via SAS for pre-prandial (-90–0 mins); each individual feeding time (0–120 mins); and fed states (120–360 mins and 360–720 mins). The proc GLIMMIX was used to analyze fed and fasted RQ and EE, and all AUC data with time (within calorimetry) and group as fixed effects and cat as subject. Group least square means (LSM) were calculated using the LSMEANS statement. The SLICEDIFF multiple comparison statement was used to compare LSM between and within the CHOLINE and CONTROL groups. Fixed effects of group, time and group*time are reported. A P<0.05 was considered significant and 0.05<P<0.10 was considered a tendency. Data is reported as LSM with the standard error of the mean (SEM).

## Results

All cats ate their assigned diet and showed no signs of illness and/or maldigestion. One cat in the CONTROL group was removed from the study and excluded from data analyses due to an event unrelated to the study.

### Daily food, calorie, and nutrient intake

Average daily intake of food, choline, and calories are presented in Fig 1, and daily protein and amino acid intake are presented in Table 2. Daily FI per kg BW was significantly lower in CHOLINE compared to CONTROL over the 12-week experimental period ($P_{Group}$<0.0001). In both groups, daily FI per kg BW changed over time ($P_{Time}$<0.0001) (Fig 1). Average daily ChI per kg BW was significantly higher in the CHOLINE group ($P_{Group}$<0.0001) and changed over time in both groups ($P_{Time}$<0.0001), with an increase in CHOLINE and decrease in CONTROL ($P_{Time}$<0.0001). Furthermore (Table 2), daily calorie and protein intake per kg BW were lower in CHOLINE compared to CONTROL ($P_{Group}$<0.0001) and decreased over time in both groups ($P_{Time}$<0.0001). All daily amino acid intakes per kg BW tended to be lower in CHOLINE compared to CONTROL ($P_{Group}$ = 0.069) and decreased over time in both groups ($P_{Time}$<0.0001).

### Body weight, gain ratios, and body composition

A significant effect of group ($P_{Group}$ = 0.017), time ($P_{Time}$ = 0.017) and group x time interaction ($P_{Group*Time}$ = 0.030) was observed for weekly BW, with CHOLINE cats experiencing a lower

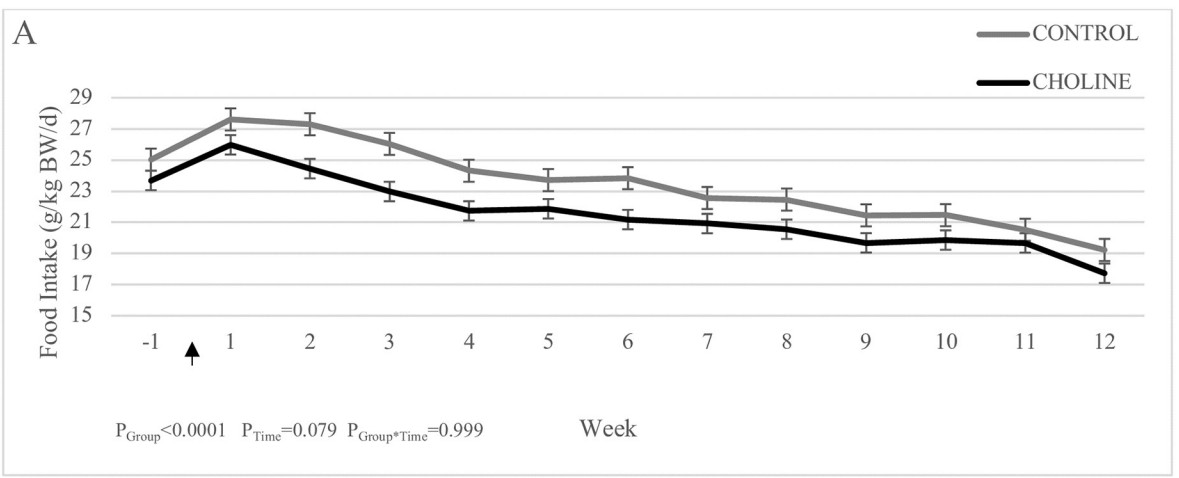

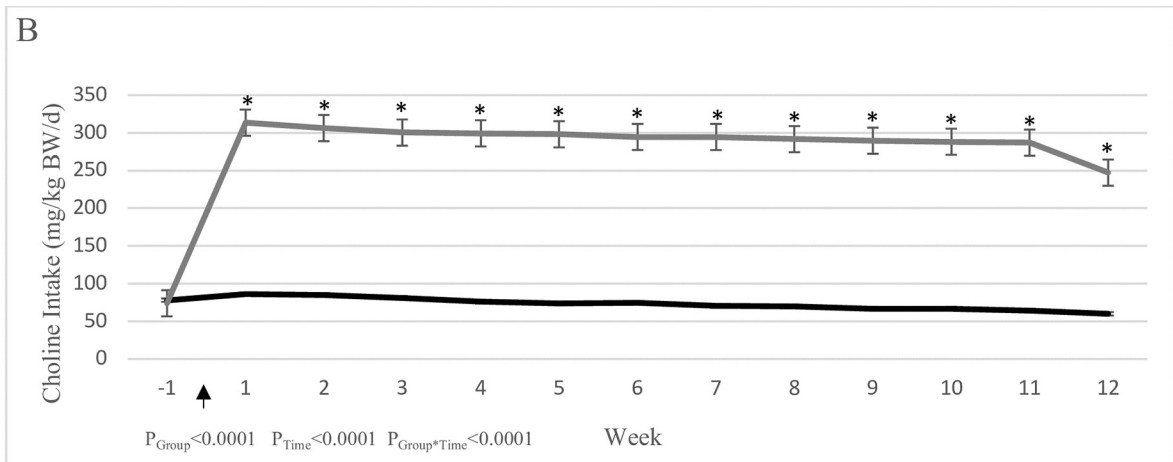

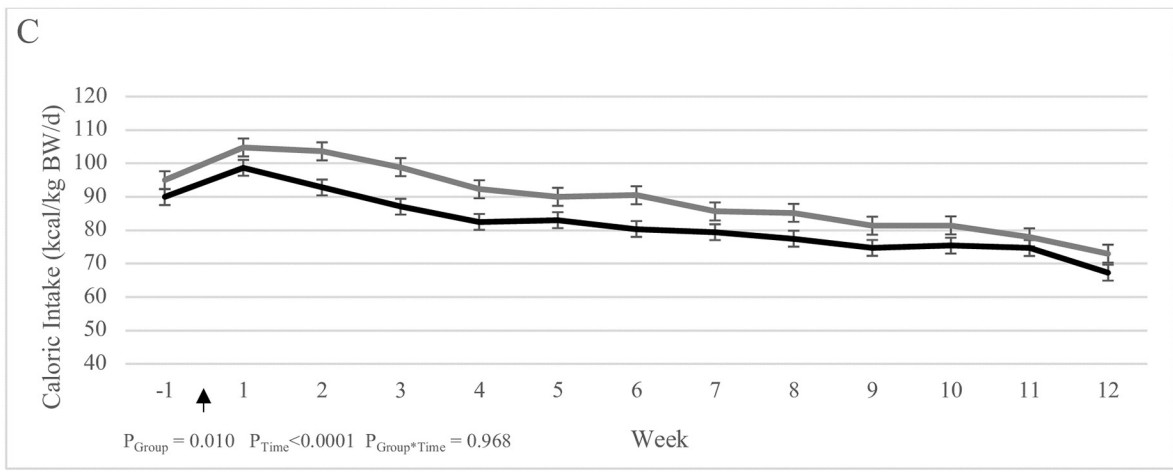

**Fig 1. Daily food intake (A), daily choline intake (B), and daily calorie intake (C) of post-gonadectomy kittens supplemented with additional choline (CHOLINE, n = 8) for 12 weeks compared to a group fed only the base diet (CONTROL, n = 7).** Gonadectomy occurred at Week 1 (arrow).

**Table 2. Mean daily protein and amino acid intakes by UPLC over 12 weeks in kittens supplemented with additional choline at 300 mg/kg BW$^{0.75}$ (CHOLINE, n = 8) compared to a control group (CONTROL, n = 7) fed only the base diet.**

| | CHOLINE (n = 8) | | CONTROL (n = 7) | | P Values | | |
|---|---|---|---|---|---|---|---|
| | LSM | SEM | LSM | SEM | $P_{Group}$ | $P_{Time}$ | $P_{Group*Time}$ |
| **Protein Intake (g/kg BW/d)** | 6.90 | 0.06 | 7.47 | 0.06 | <0.0001 | <0.0001 | 0.989 |
| **Alanine (g/kg BW/d)** | 0.45 | 0.01 | 0.48 | 0.01 | <0.0001 | <0.0001 | 0.996 |
| **Arginine (g/kg BW/d)** | 0.48 | 0.01 | 0.51 | 0.01 | <0.0001 | <0.0001 | 0.994 |
| **Aspartic Acid (g/kg BW/d)** | 0.61 | 0.02 | 0.66 | 0.02 | <0.0001 | <0.0001 | 0.996 |
| **Glutamine (g/kg BW/d)** | 0.96 | 0.03 | 1.04 | 0.03 | <0.0001 | <0.0001 | 0.996 |
| **Glycine (g/kg BW/d)** | 0.67 | 0.02 | 0.72 | 0.02 | <0.0001 | <0.0001 | 0.996 |
| **Histidine (g/kg BW/d)** | 0.16 | 0.005 | 0.17 | 0.005 | <0.0001 | <0.0001 | 0.996 |
| **Isoleucine (g/kg BW/d)** | 0.27 | 0.008 | 0.29 | 0.009 | <0.0001 | <0.0001 | 0.996 |
| **Leucine (g/kg BW/d)** | 0.50 | 0.02 | 0.54 | 0.02 | <0.0001 | <0.0001 | 0.996 |
| **Lysine (g/kg BW/d)** | 0.41 | 0.01 | 0.44 | 0.01 | <0.0001 | <0.0001 | 0.996 |
| **Phenylalanine (g/kg BW/d)** | 0.31 | 0.01 | 0.33 | 0.01 | <0.0001 | <0.0001 | 0.989 |
| **Proline (g/kg BW/d)** | 0.49 | 0.01 | 0.53 | 0.02 | <0.0001 | <0.0001 | 0.996 |
| **Serine (g/kg BW/d)** | 0.29 | 0.01 | 0.32 | 0.01 | <0.0001 | <0.0001 | 0.996 |
| **Threonine (g/kg BW/d)** | 0.27 | 0.01 | 0.29 | 0.01 | <0.0001 | <0.0001 | 0.996 |
| **Tyrosine (g/kg BW/d)** | 0.23 | 0.01 | 0.25 | 0.01 | <0.0001 | <0.0001 | 0.996 |
| **Valine (g/kg BW/d)** | 0.31 | 0.01 | 0.34 | 0.01 | <0.0001 | <0.0001 | 0.996 |
| **Methionine (g/kg BW/d)** | 0.15 | 0.004 | 0.16 | 0.005 | <0.0001 | <0.0001 | 0.996 |
| **Cysteine (g/kg BW/d)** | 0.25 | 0.008 | 0.28 | 0.008 | <0.0001 | <0.0001 | 0.996 |
| **Tryptophan (g/kg BW/d)** | 0.023 | 0.0001 | 0.025 | 0.0007 | <0.0001 | <0.0001 | 0.996 |
| **Taurine (g/kg BW/d)** | 0.046 | 0.001 | 0.049 | 0.001 | <0.0001 | <0.0001 | 0.996 |

LSM, least square means; SEM, standard error of the mean

BW gain compared to CONTROL (Fig 2A). While there was no effect of group on Gain:Feed ($P_{Group}$ = 0.324), the ratio changed over time ($P_{Time}$<0.0001) in both groups, with lower weekly Gain:Feed ratios over the 12 weeks for CHOLINE ($P_{Group*Time}$ = 0.012) (Fig 2B). The Gain:Choline ratios were significantly lower in the CHOLINE group ($P_{Group}$<0.0001) (Fig 2C). Regardless of treatment, Gain:Choline ratios for both groups changed over time ($P_{Time}$<0.0001).

Body composition from DXA analyses is summarized in Table 3. CHOLINE had lower TTM ($P_{Group}$ = 0.042) and FM ($P_{Group}$ = 0.006). No group effect was observed for LSTM, BMC and BMD ($P_{Group}$>0.05). Total tissue mass, FM, LSTM, BMC, and BMD increased over time in both groups ($P_{Time}$<0.0001); an increase in TTM ($P_{Group*Time}$ = 0.031), fat mass ($P_{Group*Time}$ = 0.005) and BMC ($P_{Group*Time}$ = 0.047) was observed in the CHOLINE group after 12-weeks of supplementation compared to CONTROL. A group x time interaction was not noted for LSTM and BMD ($P_{Group*Time}$ = 0.441 and $P_{Group*Time}$ = 0.644).

## Serum glucose, lipid profile, and liver enzymes

The mean fasted serum concentrations of glucose, lipid profile; NEFA, TAG, LDL, HDL, and HDL-C, and liver enzymes; ALT, ALP, and GGT at week -1 and week 12 for both CHOLINE and CONTROL are summarized in Table 4. There were no differences in serum concentrations between groups for all variables under fasted conditions ($P_{Groupe}$>0.05). While cholesterol, glucose, NEFA, TAG, and VLDL did not change over time, LDL and ALP decreased after 12 weeks ($P_{Time}$<0.0001) and HDL-C and ALT increased after 12 weeks ($P_{Time}$ = 0.002

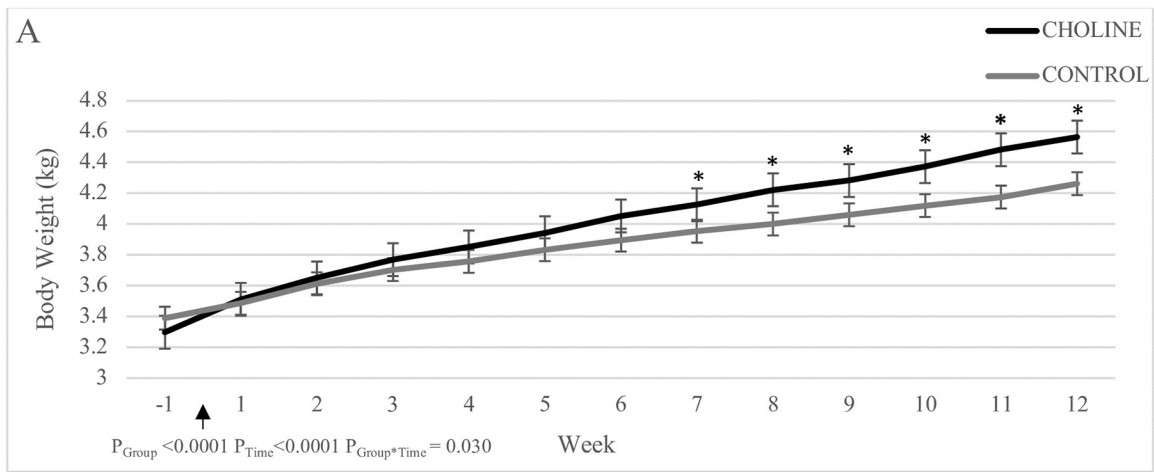

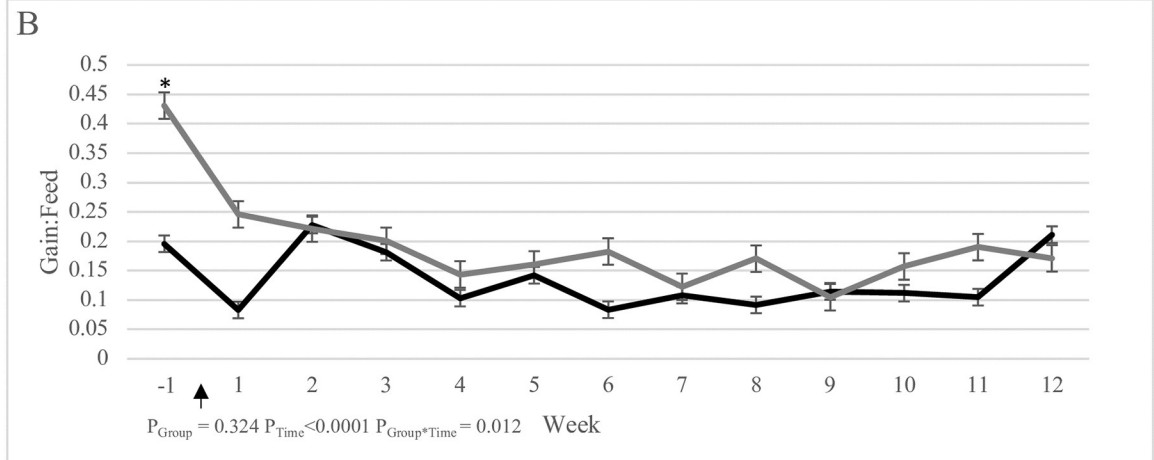

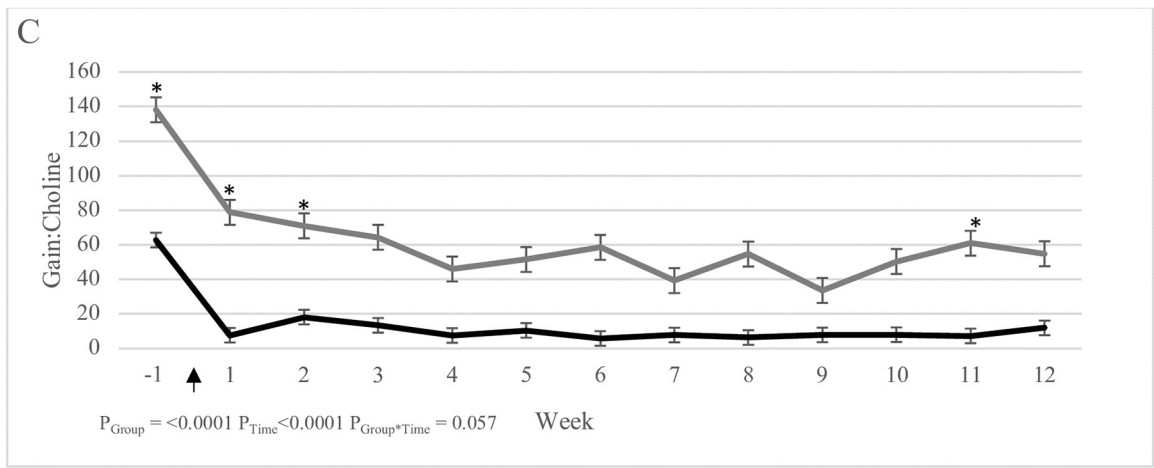

**Fig 2. Weekly body weight (A), weekly Gain:Feed ratio (B) and weekly Gain:Choline ratio (C) of post-gonadectomy kittens supplemented with additional choline (CHOLINE, n = 8) for 12 weeks compared to a group fed only the base diet (CONTROL, n = 7).** Gonadectomy occurred at week 1 (arrow).

**Table 3. Dual-energy X-ray absorptiometry measurements of post-gonadectomy kittens supplemented with additional choline at 300 mg/kg BW$^{0.75}$ (CHOLINE, n = 8) over 12 weeks compared to a group fed only the base diet (CONTROL, n = 7).**

| | Week | CHOLINE (n = 8) | | CONTROL (n = 7) | | P-Values | | |
|---|---|---|---|---|---|---|---|---|
| | | LSM | SEM | LSM | SEM | $P_{Group}$ | $P_{Time}$ | $P_{Group*Time}$ |
| TTM (g) | -1 | 3616.78 | 80.70 | 3602.12 | 86.29 | 0.042 | <0.0001 | 0.031 |
| | 12 | 4547.99 | | 4940.48 | | | | |
| FM (g) | -1 | 879.83 | 44.08 | 878.34 | 47.13 | 0.006 | <0.0001 | 0.005 |
| | 12 | 1364.53 | | 1673.39 | | | | |
| LSTM (g) | -1 | 2734.59 | 48.96 | 2734.99 | 52.34 | 0.439 | <0.0001 | 0.441 |
| | 12 | 3180.97 | | 3261.80 | | | | |
| BMC (g) | -1 | 117.98 | 2.06 | 117.96 | 2.20 | 0.051 | <0.0001 | 0.047 |
| | 12 | 150.64 | | 159.93 | | | | |
| BMD (g/cm$^2$) | -1 | 0.383 | 0.005 | 0.384 | 0.005 | 0.710 | <0.0001 | 0.644 |
| | 12 | 0.428 | | 0.424 | | | | |

BMC, bone mineral content; BMD, bone mineral density; FM, fat mass; LSTM, lean soft tissue mass; LSM, least square means; SEM, standard error of the mean; TTM, total tissue mass

and 0.011 respectively) in post-gonadectomy kittens regardless of treatment. A group x time interaction was not observed for any of the serum parameters analysed ($P_{Group*Time}$>0.05). Effect of gonadectomy and choline were not stated for GGT as it was below detection limit in all cats at week -1 and 12.

**Table 4. Fasted serum concentrations of glucose, cholesterol, NEFA, TAG, LDL, VLDL, HDL-C, ALT, and ALP of post-gonadectomy kittens supplemented with additional choline at 300 mg/kg BW$^{0.75}$ (CHOLINE, n = 8) compared to a group fed only the base diet (CONTROL, n = 7) over 12 weeks.**

| | Week | CHOLINE (n = 8) | | CONTROL (n = 7) | | P-Values | | |
|---|---|---|---|---|---|---|---|---|
| | | LSM | SEM | LSM | SEM | $P_{Group}$ | $P_{Time}$ | $P_{Group*Time}$ |
| Glucose (mmol/L) | -1 | 15.68 | 0.69 | 15.85 | 0.73 | 0.491 | 0.140 | 0.639 |
| | 12 | 14.22 | | 15.08 | | | | |
| Cholesterol (mmol/L) | -1 | 4.36 | 0.18 | 4.42 | 0.19 | 0.607 | 0.716 | 0.395 |
| | 12 | 4.45 | | 4.20 | | | | |
| NEFA (mmol/L) | -1 | 0.31 | 0.03 | 0.29 | 0.03 | 0.174 | 0.715 | 0.091 |
| | 12 | 0.26 | | 0.37 | | | | |
| TAG (mmol/L) | -1 | 0.39 | 0.04 | 0.40 | 0.04 | 0.156 | 0.134 | 0.480 |
| | 12 | 0.43 | | 0.50 | | | | |
| LDL (mmol/L) | -1 | 1.03 | 0.08 | 1.10 | 0.08 | 0.181 | <0.0001 | 0.151 |
| | 12 | 0.60 | | 0.37 | | | | |
| VLDL (mmol/L) | -1 | 0.18 | 0.02 | 0.18 | 0.02 | 0.153 | 0.133 | 0.476 |
| | 12 | 0.19 | | 0.23 | | | | |
| HDL-C (mmol/L) | -1 | 3.13 | 0.17 | 3.13 | 0.18 | 0.209 | 0.002 | 0.212 |
| | 12 | 4.06 | | 3.59 | | | | |
| ALT (U/L) | -1 | 37.0 | 4.09 | 35.0 | 4.37 | 0.373 | 0.011 | 0.167 |
| | 12 | 43.63 | | 53.43 | | | | |
| ALP (U/L) | -1 | 95.0 | 8.55 | 100.14 | 9.14 | 0.580 | <0.0001 | 0.726 |
| | 12 | 53.50 | | 61.71 | | | | |

ALT, Alanine Aminotransferase; ALP, alkaline phosphatase; HDL-C, high-density lipoprotein cholesterol; LDL, low-density lipoprotein; LSM, least square means; NEFA, non-esterified fatty acid; SEM, standard error of the mean; TAG, triacylglyceride; VLDL, very low-density lipoprotein

**Table 5. Fasted serum satiety hormone concentrations of post-gonadectomy kittens supplemented with additional choline at 300 mg/kg BW$^{0.75}$ (CHOLINE, n = 8) compared to a group fed only the base diet (CONTROL, n = 7) over 12 weeks.**

| | Week | CHOLINE (n = 8) | | CONTROL (n = 7) | | P Values | | |
|---|---|---|---|---|---|---|---|---|
| | | LSM | SEM | LSM | SEM | P$_{Group}$ | P$_{Time}$ | P$_{Group*Time}$ |
| **Leptin (ng/mL)** | -1 | 4.04 | 0.32 | 3.77 | 0.34 | 0.946 | 0.141 | 0.467 |
| | 12 | 3.28 | | 3.50 | | | | |
| **Ghrelin (ng/mL)** | -1 | 15.13 | 2.58 | 14.19 | 2.76 | 0.804 | 0.512 | 0.924 |
| | 12 | 16.67 | | 16.24 | | | | |
| **GLP-1 (ng/mL)** | -1 | 1.03 | 0.42 | 1.08 | 0.50 | 0.703 | 0.981 | 0.765 |
| | 12 | 0.90 | | 1.21 | | | | |
| **PYY (ng/mL)** | -1 | 2.48 | 0.51 | 1.71 | 0.61 | 0.983 | 0.810 | 0.179 |
| | 12 | 1.56 | | 2.36 | | | | |
| **GIP (ng/mL)** | -1 | 286.34 | 16.67 | 277.05 | 17.84 | 0.220 | 0.318 | 0.082 |
| | 12 | 271.43 | | 328.26 | | | | |

GLP-1, glucagon-like peptide-1; GIP, gastric inhibitory polypeptide; LSM, least square means; PYY, Peptide-YY; SEM, standard error of the mean

## Satiety hormones

Mean fasted serum concentrations of leptin, ghrelin, GLP-1, PYY, and GIP at week -1 and week 12 for CHOLINE and CONTROL are displayed in Table 5. There were no significant group effects on any of the satiety hormones assessed (P$_{Group}$>0.05). Similarly, there was no effect of time and no group*time interaction was found for satiety hormones (P$_{Time}$>0.05; P$_{Group*Time}$>0.05).

## Energy expenditure and respiratory quotient

As shown in Table 6, there were no differences found between CHOLINE and CONTROL in either EE or RQ, in the fed or fasted states (P$_{Group}$>0.05). Twelve-weeks post-gonadectomy resulted in a decrease in EE in the fed state regardless of treatment (P$_{Time}$ = 0.004). When adjusted for FI or BW, this decrease was still prevalent (P$_{Time}$ = 0.011 and P$_{Time}$ = 0.049 respectively). In the fasted state, RQ was greater 12-weeks post-gonadectomy (P$_{Time}$ = 0.014). When adjusted for FI, the increase in RQ was still present (P$_{Time}$ = 0.011). When adjusted for BW, there was no significant effect of time (P$_{Time}$ = 0.908). Area under the curve for RQ was not different between groups (P$_{Group}$>0.05) but increased at 12-weeks in post-gonadectomy regardless of treatment from 0–120 mins during the first feeding (P$_{Time}$<0.0001). Similarly, there was a decrease in AUC from 120–360 mins in the post-fed state after 12 weeks, again regardless of treatment (P$_{Time}$ = 0.005). No group x time interactions were observed for EE or RQ in the fed or fasted states (P$_{Group*Time}$>0.05).

## Discussion

It is generally accepted that choline is an essential nutrient for mammals, including cats. Studies evaluating choline deficiency in kittens have demonstrated a need for choline in the diet during growth [16, 17]. These contribute to the current recommendation for choline intake of 133 mg/kg BW$^{0.67}$, and the minimum requirement of 107 mg/kg BW$^{0.67}$; however, these recommendations were based on a dearth of data. On average, kittens fed CHOLINE were consuming 3.2 times the recommended daily allowance, and 4.0 times the minimum requirement. In comparison, the CONTROL group consumed an average of 0.8 times the recommended daily allowance, and 1.0 times the minimum requirement.

**Table 6. Energy expenditure, respiratory quotient, and area under the curve for respiratory quotient of post-gonadectomy kittens supplemented with additional choline at 300 mg/kg $BW^{0.75}$ (CHOLINE, n = 8) for 12 weeks compared to a group fed only the base diet (CONTROL, n = 7).**

| | Week | CHOLINE (n = 8) | | CONTROL (n = 7) | | P-Values | | |
|---|---|---|---|---|---|---|---|---|
| | | LSM | SEM | LSM | SEM | $P_{Group}$ | $P_{Time}$ | $P_{Group*Time}$ |
| **EE Fasted (kcal/kg BW)** | -1 | 25.50 | 2.47 | 25.45 | 2.65 | 0.114 | 0.123 | 0.681 |
| | 12 | 21.19 | | 18.18 | | | | |
| Adjusted for FI EE Fasted § | -1 | 25.56 | 3.06 | 25.47 | 3.28 | 0.626 | 0.092 | 0.644 |
| | 12 | 21.17 | | 18.10 | | | | |
| Adjusted for BW EE Fasted ‡ | -1 | 25.10 | 4.06 | 24.97 | 5.04 | 0.671 | 0.461 | 0.704 |
| | 12 | 21.49 | | 18.77 | | | | |
| **EE Fed (kcal/kg BW)** | -1 | 28.75 | 2.32 | 31.15 | 2.48 | 0.976 | 0.004 | 0.453 |
| | 12 | 20.33 | | 17.84 | | | | |
| Adjusted for FI EE Fed § | -1 | 26.68 | 3.05 | 31.36 | 3.27 | 0.541 | 0.011 | 0.407 |
| | 12 | 19.75 | | 19.02 | | | | |
| Adjusted for BW EE Fed ‡ | -1 | 28.94 | 4.15 | 33.86 | 5.16 | 0.715 | 0.049 | 0.304 |
| | 12 | 18.17 | | 15.75 | | | | |
| **RQ Fasted** | -1 | 0.75 | 0.01 | 0.75 | 0.01 | 0.664 | 0.014 | 0.856 |
| | 12 | 0.78 | | 0.78 | | | | |
| Adjusted for FI RQ Fasted § | -1 | 0.74 | 0.01 | 0.76 | 0.01 | 0.198 | 0.011 | 0.923 |
| | 12 | 0.77 | | 0.79 | | | | |
| Adjusted for BW RQ Fasted ‡ | -1 | 0.76 | 0.01 | 0.78 | 0.01 | 0.432 | 0.908 | 0.468 |
| | 12 | 0.76 | | 0.77 | | | | |
| **RQ Fed** | -1 | 0.82 | 0.01 | 0.82 | 0.01 | 0.425 | 0.564 | 0.516 |
| | 12 | 0.82 | | 0.83 | | | | |
| Adjusted for FI RQ Fed § | -1 | 0.82 | 0.01 | 0.82 | 0.01 | 0.284 | 0.208 | 0.464 |
| | 12 | 0.82 | | 0.84 | | | | |
| Adjusted for BW RQ Fed ‡ | -1 | 0.82 | 0.01 | 0.82 | 0.01 | 0.688 | 0.245 | 0.911 |
| | 12 | 0.82 | | 0.83 | | | | |
| **$AUC_{RQ}$ Pre-Prandial (RQ*min)** | -1 | 43.45 | 2.25 | 42.12 | 2.10 | 0.388 | 0.733 | 0.148 |
| | 12 | 39.38 | | 44.69 | | | | |
| **$AUC_{RQ}$ Feeding Time 1 (0–120 min) (RQ*min)** | -1 | 91.21[c] | 0.98 | 91.48 | 0.92 | 0.965 | <0.0001 | 0.817 |
| | 12 | 96.84[d] | | 96.66 | | | | |
| **$AUC_{RQ}$ Feeding Time 2 (0–120 min) (RQ*min)** | -1 | 95.52 | 2.57 | 97.50 | 2.40 | 0.282 | 0.450 | 0.710 |
| | 12 | 96.50 | | 100.35 | | | | |
| **$AUC_{RQ}$ Feeding Time 3 (0–120 min) (RQ*min)** | -1 | 109.96 | 6.48 | 102.39 | 5.71 | 0.732 | 0.190 | 0.125 |
| | 12 | 92.78 | | 103.87 | | | | |
| **$AUC_{RQ}$ 120–360 min (RQ*min)** | -1 | 183.41 | 2.73 | 183.93 | 2.73 | 0.234 | 0.005 | 0.303 |
| | 12 | 170.92 | | 177.51 | | | | |
| **$AUC_{RQ}$ 360–720 min (RQ*min)** | -1 | 274.18 | 10.08 | 274.30 | 9.43 | 0.821 | 0.971 | 0.812 |
| | 12 | 276.92 | | 271.30 | | | | |

$AUC_{RQ}$, area under the curve for respiratory quotient; BW, body weight; EE, energy expenditure; FI, food intake; LSM, least square means; RQ, respiratory quotient; SEM, standard error of the mean

§ Food intake at time of calorimetry used as a covariate

‡ Body weight at time of calorimetry used as a covariate

In the present study, gonadectomized kittens were fed to mimic *ad libitum* feeding to determine if additional choline decreased daily FI similar to previous reports in growing swine [27] and poultry [48]. Additional choline resulted in post-gonadectomy kittens consuming 11% less food than the CONTROL group. As a result, daily energy intake for the CHOLINE group

was also lower compared to CONTROL. Mean energy intake was 2.7% and 18% above the calculated DER for growth [19] for CHOLINE and CONTROL, respectively. Interestingly, although the gonadectomized kittens in the present study were consuming above the NRC recommendations when offered *ad libitum* feeding, daily FI increased significantly by 14% and 10% in the CONTROL and CHOLINE groups, respectively, for the first week immediately following gonadectomy, followed by a decrease over the rest of the study duration in both groups. These findings are consistent with the findings by Wei et al. [9] in which immediately following neuter, cats experienced a period of hyperphagia.

The lower daily FI coincides with the lower weekly BW gain observed in the CHOLINE group in the present study. Previous reports by Fettman et al. [49] show that the increase in FI is the most likely cause for the increased BW in neutered cats, although it could also be due to a reduction in energy requirements that has previously been hypothesized as a result of gonadectomy, and/or reduced EE or voluntary physical activity [8–10]. In the present study, kittens were expected to have an increase in BW (38) unlike previous studies in adult cats [9]. Indeed, all kittens increased BW throughout the study, and choline appeared to have an effect on reducing the amount of BW gain, likely due to reduced FI; a 26% increase in BW from preneuter to 12-weeks post-neuter was seen for CHOLINE compared to a 38% increase in BW for CONTROL.

Interestingly, Matthews et al. [27] found that average daily gain was not affected when pigs were supplemented with betaine. These pigs continued to grow similarly throughout the study, whereas the CONTROL group in the present study gained more than those in the CHOLINE. In pigs, changes in carcass composition were observed, including a 10.5% reduction in percentage carcass fat, and increased daily gain of carcass lean mass [27]. Total tissue mass by DXA in the kittens was 7.9% lower in the CHOLINE group at week 12 than CONTROL. Similar to Matthews et al. [27], the kittens fed CHOLINE had 18.5% less FM than the CONTROL group at week 12. In contrast to the effects of betaine supplementation in pigs, LSTM gain was not higher with CHOLINE. While the CHOLINE group consumed, on average, 12% less protein than the CONTROL group, both groups were consuming above the minimum protein and amino acid requirements for growing kittens recommended by the NRC [19]. Methionine intake specifically was consumed at 37.5% and 50% greater than the NRC [19] minimum requirement by the CHOLINE and CONTROL groups, respectively. The fact that LSTM was not affected, although the CHOLINE group consumed less protein, could be indicative of higher protein deposition with additional choline, likely due to its involvement in one-carbon metabolism in generating methionine from homocysteine and ultimately increasing concentrations of SAM [25]. However, since both groups were consuming amino acids and protein above the NRC [19] requirements for growth, it is likely that both groups consumed sufficient amounts for maximal growth. Overall, growth of the kittens in the present study was as expected and BW trends were in line with, or above, published kitten growth charts [38, 50], and were within or above the ranges in the cross-sectional study by Lauten et al. [51] for kittens between 6 to 10 months of age.

In the present study, additional choline did not alter serum glucose, cholesterol, NEFA, TAG and lipoproteins under fasted conditions and serum concentrations of cholesterol and TAG were similar to previously published concentrations of healthy adult cats [52]. Serum concentrations of glucose were above reference ranges, likely due to sedation and stress during collection [53]. In choline deficient mice models, VLDL and LDL components were decreased by half [54]. When dietary choline was then supplemented, VLDL and LDL were restored to similar levels of the control group. In rats, PC increased VLDL exportation, however, did not affect HDL-C from hepatic tissue [55]. Previous studies demonstrating this effect of choline on lipoproteins are typically in choline deficient rodent models, rodent models with non-

alcoholic fatty liver disease, or hepatic steatosis, or fed high fat diets. Rodent models can provide insights into the mechanisms; however, caution should be considered when comparing to feline research due to nutritional and metabolic differences between species [56]. In the present study, due to the mimicked *ad libitum* feeding design of the study, the two groups consumed different amounts of food, energy sources, and lipids with differing choline amounts between groups. A study that controls dietary intake (g/kg BW$^{0.75}$/d) may provide further mechanistic understanding into the effects of choline supplementation in kittens.

Since kittens were in the growth phase and had a normal body condition, they would not have adipose tissue stores that could impact VLDL secretion from the liver, as the majority of TAG in VLDL is from lipid droplets that have been mobilized to the liver [20]. Therefore, kittens will not have abundant hepatic storage of fatty acids as TAG to be packaged and exported in VLDL. However, adult obese cats, which have higher hepatic TAG stores than lean cats [57], supplemented with choline would have had prominent hepatic stores. This higher hepatic storage may require an increase in VLDL mobilization with PC. Therefore, this may account for the observed increase in VLDL levels in obese adult cats in the study by Verbrugghe et al. [24].

Kittens in both groups had fasted RQ levels indicative of fatty acid oxidation [58]. Fasted RQ levels tend to be lower, indicative of greater proportions of fatty acid oxidation, than RQ in the fed state, and this was also found in pigs [59]. This pattern of RQ fluctuation was further confirmed in the fed state where the RQ appeared to increase, suggesting that kittens in the fed state relied on a mix of macronutrients for energy [58, 60]. Choline has previously been shown to increase muscle carnitine concentrations, and thus increase fatty acid oxidation in guinea pigs [31] as well as in rats [61]. Results from indirect calorimetry respiratory quotients in the present study do not suggest that this occurred in the growing kittens, although muscle carnitine concentrations were not measured directly in these kittens.

Gonadectomy may have impacted EE under the fed state. Energy expenditure was decreased in both groups after 12 weeks post-gonadectomy, regardless of dietary choline intake and suggests that gonadectomy could result in lower energy requirements similar to previous research in spayed female cats [62]. While Flynn et al. [62] did not assess EE via indirect calorimetry, they found that the maintenance energy requirements to maintain an ideal BW decreased by 24–30% in spayed female cats. Using the doubly labelled water method to determine EE, Martin et al. [8], also found that EE decreased after neuter or spay in both male and female cats. However, without an intact group to compare, this cannot be confirmed.

In the present study, there was no difference in resting EE between CHOLINE and CONTROL, although it was thought that choline supplementation may increase EE and therefore minimize the effects of gonadectomy on EE. This is because choline supplementation was thought to increase lean mass, as shown in other species [27, 63]. Robinson et al. [64] demonstrated the importance of choline, betaine, and methionine on protein deposition and muscle synthesis via one-carbon metabolism and further supports the idea that LSTM would be impacted and therefore affect EE as lean mass is a key determinant on EE due to its higher energy needs for maintenance [65]. In the present study, however, LSTM was not different from the CONTROL group, only FM, and FM is not associated with large changes in EE [65].

Satiety hormones were also assessed in the present study. Gonadectomy results in higher FI and hyperphagia [9], and supplementation of choline in the present study lowers daily FI compared to non-supplemented kittens, post-gonadectomy. Overall, satiety hormones have not been well studied in response to either gonadectomy, age of gonadectomy, or choline supplementation, specifically in cats. In the present study, choline supplementation did not alter fasted serum levels of any of these hormones. Thereby, we cannot attribute any of the significant findings in FI, BW, TTM, BMC, or FM to satiety. Similar to the conundrum of the lipoproteins, since FI was not controlled for in the present study, this may have impacted the

satiety hormone results. Satiety hormones were also only assessed in the fasted state. Evaluating the post-prandial changes in satiety hormones may provide a more in depth understanding as to whether choline supplementation impacts serum satiety hormone concentrations. Due to its cholinergic effects as acetylcholine, this may be an area that should be further investigated as this could have a direct impact on satiety hormones. Furthermore, understanding the relationship between choline, FM, FI, and serum leptin and ghrelin needs to be further investigated.

Betaine and L-carnitine supplements have been more thoroughly investigated for improved growth as well for obesity prevention and treatments. A study in cats comparing the efficacy of choline to its derivatives and/or investigating combination dosing may also be warranted. L-carnitine has been documented to provide beneficial effects in various models. For example, L-carnitine supplementation showed FI lowering effects and impacts on EE which were more pronounced in obese cats versus lean cats [66, 67]. This may have implications in the present study as there was variability in BW and BCS due to the *ad libitum* FI and growth. Similarly, choline prevented cirrhosis of the liver in rats fed a high fat diet [68] and its lipotropic effects were again, more pronounced with a high fat diet than a low-fat diet in black seabream [69]. Therefore, the lipotropic effects of choline supplementation may be positively associated with body FM and dietary fat intake.

Based on the findings of the present study, choline supplementation may have a protective effect against the obesity related effects of gonadectomy and *ad libitum* feeding in kittens during growth. However, it is unclear how additional choline works to lower FI, BW, and FM. Future studies should aim to assess choline supplementation during a controlled FI period. While the current study used a dose based on previous research of a dose response study of choline in rats [39] and in growing pigs [40], as well as a preliminary study in obese cats [24], a thorough dose response study in growing kittens and adult cats is also needed. Current choline requirements for kittens and adult cats were established via choline deficiency studies, not empirical measures of the requirement using a dose response approach, and evaluating liver lipid content [16, 18]. The current study used different outcome measures, suggesting that choline supplementation above previously accepted requirements may have additional health benefits in cats. This calls for further research on supplementation of choline and its derivates in cats to ensure optimal health and proper growth, as well as obesity prevention and treatment.

## Acknowledgments

A special thank you to Michelle Beaudoin-Kimble for performing the serum satiety hormone analyses, Cara Cargo-Froom for helping with the diet amino acid analyses and to Michelle Edwards for statistical support. Additionally, thank you to the Ontario Veterinary College Pet Nutrition Team and student volunteers for help with study preparation, sample collections and DXA scans as well as animal handling and enrichment throughout the study.

## Author Contributions

**Conceptualization:** Marica Bakovic, Adronie Verbrugghe.

**Data curation:** Hannah Godfrey, Anna Kate Shoveller, Adronie Verbrugghe.

**Formal analysis:** Hannah Godfrey, Anna Kate Shoveller, Marica Bakovic, Sarah K. Abood, Adronie Verbrugghe.

**Funding acquisition:** Marica Bakovic, Adronie Verbrugghe.

**Investigation:** Hannah Godfrey, Alexandra Rankovic, Caitlin E. Grant.

**Methodology:** Hannah Godfrey, Alexandra Rankovic, Caitlin E. Grant, Anna Kate Shoveller, Marica Bakovic, Sarah K. Abood, Adronie Verbrugghe.

**Project administration:** Hannah Godfrey, Anna Kate Shoveller, Adronie Verbrugghe.

**Resources:** Anna Kate Shoveller, Adronie Verbrugghe.

**Software:** Hannah Godfrey, Anna Kate Shoveller, Adronie Verbrugghe.

**Supervision:** Anna Kate Shoveller, Marica Bakovic, Sarah K. Abood, Adronie Verbrugghe.

**Validation:** Hannah Godfrey, Anna Kate Shoveller, Marica Bakovic, Sarah K. Abood, Adronie Verbrugghe.

**Visualization:** Hannah Godfrey, Anna Kate Shoveller, Marica Bakovic, Sarah K. Abood, Adronie Verbrugghe.

**Writing – original draft:** Hannah Godfrey.

**Writing – review & editing:** Hannah Godfrey, Alexandra Rankovic, Caitlin E. Grant, Anna Kate Shoveller, Marica Bakovic, Sarah K. Abood, Adronie Verbrugghe.

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
