## [Decision Letter · Decision Letter 0]

4 Jan 2022

PONE-D-21-36602Dietary choline in gonadectomized kittens improved food intake and body composition but not satiety, serum lipids, or energy expenditurePLOS ONE

Dear Dr. Verbrugghe,

Thank you for submitting your manuscript to PLOS ONE. After careful consideration, we feel that it has merit but does not fully meet PLOS ONE’s publication criteria as it currently stands. Therefore, we invite you to submit a revised version of the manuscript that addresses the points raised during the review process.

We look forward to receiving your revised manuscript.

Kind regards,

Ewa Tomaszewska, DVM Ph.D

Academic Editor

PLOS ONE

Journal Requirements:

2.  To comply with PLOS ONE submissions requirements, please provide methods of sacrifice in the Methods section of your manuscript.'

3. Thank you for stating the following in the Competing Interests:

(The authors declare no conflicts of interest. AV is the Royal Canin Veterinary Diets Endowed Chair in Canine and Feline Clinical Nutrition and declares that they serve on the Health and Nutrition Advisory Board for Vetdiet. AV has received honoraria and research funding from various pet food manufacturers and ingredient suppliers. HG declares that they have paid engagements with pet food companies within Canada. AR declares that they have participated in paid internships and engagements with pet food companies within Canada. CG holds the Nestlé Purina Professorship in Companion Animal Nutrition at the Ontario Veterinary College, is the owner of Grant Veterinary Nutrition Services and consults with Simmons Pet Food. The Ontario Veterinary College received funding from Nestlé Purina Proplan Veterinary Diets to support a Registered Veterinary Technician in Clinical Nutrition, who assisted with this study. AKS was previously employed by P&G and Mars Pet Care, serves on the Scientific Advisory Board for Trouw Nutrition, and has received honoraria and research funding from various commodity groups, pet food manufacturers, and ingredient suppliers. SKA is the owner of Sit, Stay Speak Nutrition LLC and provides nutrition consultation to industry partners.This does not alter our adherence to PLOS ONE policies on sharing data and materials.)

We note that one or more of the authors are employed by a commercial company: (Nestlé Purina Professorship, Grant Veterinary Nutrition Services, Simmons Pet Food, from Nestlé Purina Proplan Veterinary Diets, P&G, Mars Pet Care, Trouw Nutrition, and  Sit, Stay Speak Nutrition LLC)

Reviewers' comments:

Reviewer's Responses to Questions

**Comments to the Author**

1. Is the manuscript technically sound, and do the data support the conclusions?

Reviewer #1: Yes

Reviewer #2: Yes

2. Has the statistical analysis been performed appropriately and rigorously? 

Reviewer #1: Yes

Reviewer #2: Yes

3. Have the authors made all data underlying the findings in their manuscript fully available?

Reviewer #1: Yes

Reviewer #2: Yes

4. Is the manuscript presented in an intelligible fashion and written in standard English?

Reviewer #1: Yes

Reviewer #2: Yes

5. Review Comments to the Author

Reviewer #1: Review of the article No. PONE-D-21-36602, entitled “Dietary choline in gonadectomized kittens improved food intake and body composition but not satiety, serum lipids, or energy expenditure”

The aim of the study was to assess the effect of dietary supplementation with choline in cats on selected parameters, which may be influenced by gonadectomy and subsequent possible obesity.

Overall, the authors have done a very good job describing their methods and analysing the data. The manuscript is well written, and results are easy to follow. It is worth to underlined, that the data obtained in the study are very important both from a dietary and physiological point of view. The discussion is very thorough and easy to read, and clearly presents the results and comparisons. The interpretations and conclusions are proved by the data.

In conclusion, excellent work!

Reviewer #2: The study is interesting. However, some minor points must be addressed before the manuscript can be considered for publication:

L112 Please use SI units.

L195 Since 2003, a recommended by International Society for Clinical Densitometry (ISCD) abbreviation for dual energy x-ray absorptiometry is DXA. Please consider changing DEXA to DXA.

L249 Please explain the abbreviations glucagon-like peptide-1 (GLP-1) and others as you do in L255-256.

L299 CO2 and O2 – subscript; Energy expenditure (EE)

L302 EE and RQ, respectively ?

L326 Name that group (CONTROL)

L333 and L335 shouldn't be " <" for Ptime ?

6. PLOS authors have the option to publish the peer review history of their article (what does this mean?). If published, this will include your full peer review and any attached files.

Reviewer #1: **Yes: **Sylwester Kowalik

Reviewer #2: No

---

## [Author Response · Author response to Decision Letter 0]

27 Jan 2022

Journal Requirements

1. When submitting your revision, we need you to address these additional requirements. Please ensure that your manuscript meets PLOS ONE's style requirements, including those for file naming. The PLOS ONE style templates can be found at

Response: We have addressed the changes required to meet the PLOS ONE journal requirements according to the PLOS ONE templates and formatting guides. All changes to the manuscript have been tracked in the track changes version.

2. To comply with PLOS ONE submissions requirements, please provide methods of sacrifice in the Methods section of your manuscript

Response: While we understand that methods of sacrifice are required in the Methods section, this manuscript does not have sacrifice the animals used in the study.

3. Please provide an amended Funding Statement declaring this commercial affiliation, as well as a statement regarding the Role of Funders in your study. If the funding organization did not play a role in the study design, data collection and analysis, decision to publish, or preparation of the manuscript and only provided financial support in the form of authors' salaries and/or research materials, please review your statements relating to the author contributions, and ensure you have specifically and accurately indicated the role(s) that these authors had in your study. You can update author roles in the Author Contributions section of the online submission form. Please also include the following statement within your amended Funding Statement. “The funder provided support in the form of salaries for authors [insert relevant initials], but did not have any additional role in the study design, data collection and analysis, decision to publish, or preparation of the manuscript. The specific roles of these authors are articulated in the ‘author contributions’ section.” If your commercial affiliation did play a role in your study, please state and explain this role within your updated Funding Statement. 

Response: The authors have reviewed the funding statement. The statement has been updated to include funding for salaries, including the stipend for the MSc. Student, Dr. Verbrugghe’s endowed chair position and funding for a technician who assisted with the study. Any other commercial affiliations listed in the competing interest statement are not related to this research. The changes are in the response below. 

“This research was funded by a Natural Sciences and Engineering Research Council (NSERC) Collaborative Research and Development grant (CRDPJ 472710-16) in partnership with Elmira Pet Products (Elmira, ON, Canada). The Choline Chloride supplement was kindly provided by Balchem (New Hampton, NY, USA). Salaries were supported by an Ontario Veterinary College Scholarship for the MSc stipend for HG, the Royal Canin Veterinary Diets Endowed Chair in Canine and Feline Clinical Nutrition for AV’s faculty position at the Ontario Veterinary College and funding from Nestlé Purina Proplan Veterinary Diets to support a Registered Veterinary Technician in Clinical Nutrition, who assisted with this study. The funders had no role in the study design, data collection and analysis, decision to publish, or preparation of the manuscript. Elmira Pet Products and Balchem reviewed the final manuscript before submission for publication. The specific roles of these authors are articulated in the ‘author contributions’ section.”

We have updated our author contributions as follows:

Conceptualization: AV, MB

Data Curation: HG, AV, AKS

Formal Analysis: HG, AV, MB, AKS, SKA

Funding acquisition: AV, MB

Methodology: HG, AV, MB, AKS, SKA, AR, CG

Project Administration: HG, AV, AKS

Resources: AV, AKS

Software: AV, AKS, HG

Visualization: HG, AV, SKA, MB, AKS

Validation: AV, SKA, MB, AKS

Investigation: HG, AR, CG

Supervision: AV, SKA, MB, AKS

Writing – Original Draft: HG

Writing – Reviewing and Editing: AR, CG, AV, SKA, MB, AKS

4. Please also provide an updated Competing Interests Statement declaring this commercial affiliation along with any other relevant declarations relating to employment, consultancy, patents, products in development, or marketed products, etc. Within your Competing Interests Statement, please confirm that this commercial affiliation does not alter your adherence to all PLOS ONE policies on sharing data and materials by including the following statement: "This does not alter our adherence to PLOS ONE policies on sharing data and materials.” (as detailed online in our guide for authors http://journals.plos.org/plosone/s/competing-interests) . If this adherence statement is not accurate and there are restrictions on sharing of data and/or materials, please state these. Please note that we cannot proceed with consideration of your article until this information has been declared.

Response: We have an updated Competing Interests Statement that declares all commercial affiliations and confirms that this does not alter our adherence to all PLOS ONE policies:

“HG declares that they had paid engagements with pet food companies within Canada. AV is the Royal Canin Veterinary Diets Endowed Chair in Canine and Feline Clinical Nutrition at the Ontario Veterinary College and declares that they serve on the Health and Nutrition Advisory Board for Vetdiet. AV has also received honoraria and research funding from various pet food manufacturers and ingredient suppliers. AR declares that they participated in paid internships and engagements with pet food companies within Canada. Upon manuscript submission, CG holds the Nestlé Purina Professorship in Companion Animal Nutrition at the Ontario Veterinary College and is the owner of Grant Veterinary Nutrition Services. CG has also consulted with Simmons Pet Food. At the time of manuscript submission, AKS, a previous employee of P&G and Mars Pet Care, holds the Champion Petfoods Chair in Canine and Feline Nutrition, Physiology and Metabolism at the Ontario Agricultural College. AKS also serves on the Scientific Advisory Board for Trouw Nutrition, and has received honoraria and research funding from various commodity groups, pet food manufacturers, and ingredient suppliers. SKA is the owner of Sit, Stay Speak Nutrition LLC and declares that they have provided nutrition consultation to industry partners. These commercial affiliations do not alter the authors’ adherence to PLOS ONE policies on sharing data and materials. “

5. In your Data Availability statement, you have not specified where the minimal data set underlying the results described in your manuscript can be found. PLOS defines a study's minimal data set as the underlying data used to reach the conclusions drawn in the manuscript and any additional data required to replicate the reported study findings in their entirety. All PLOS journals require that the minimal data set be made fully available. For more information about our data policy, please see http://journals.plos.org/plosone/s/data-availability. Upon re-submitting your revised manuscript, please upload your study’s minimal underlying data set as either Supporting Information files or to a stable, public repository and include the relevant URLs, DOIs, or accession numbers within your revised cover letter. For a list of acceptable repositories, please see http://journals.plos.org/plosone/s/data-availability#loc-recommended-repositories. Any potentially identifying patient information must be fully anonymized.

Response: For our data availability, the minimal data sets are available from the Scholars Portal Dataverse for the University of Guelph (https://doi.org/10.5683/SP3/OJZEVR). This DOI has been reserved for the dataset but will not become an active link until the dataset is published and publicly available in the repository. The temporary link for the dataset for coauthors, reviewers, and publishers is provided below:

https://dataverse.scholarsportal.info/dataset.xhtml;jsessionid=469a86bcb473b6ba0a9fab371a85?persistentId=doi%3A10.5683%2FSP3%2FOJZEVR&version=DRAFT

Response: All edits to the reference list have been tracked in the revised manuscript. No citations have been removed or added, only updated to fit the required formatting for PLOS ONE. There are no cited articles that have been retracted to our knowledge.

Reviewer 1

The aim of the study was to assess the effect of dietary supplementation with choline in cats on selected parameters, which may be influenced by gonadectomy and subsequent possible obesity.

Overall, the authors have done a very good job describing their methods and analysing the data. The manuscript is well written, and results are easy to follow. It is worth to underlined, that the data obtained in the study are very important both from a dietary and physiological point of view. The discussion is very thorough and easy to read, and clearly presents the results and comparisons. The interpretations and conclusions are proved by the data.

In conclusion, excellent work!

Response: We are so grateful for these kind words and would like to extend our thanks to reviewer 1. 

Reviewer 2

The study is interesting. However, some minor points must be addressed before the manuscript can be considered for publication:

Response: We thank reviewer 2 for their kind words and have responded to their suggestions below. 

7. L112 Please use SI units: 

Response: Thank you for identifying this error. We have changed “ft” to “m” to ensure SI units.

8. L195 Since 2003, a recommended by International Society for Clinical Densitometry (ISCD) abbreviation for dual energy x-ray absorptiometry is DXA. Please consider changing DEXA to DXA: 

Response: We have changed our abbreviation for Dual Energy X-Ray Absorptiometry to DXA to suit the ISCD recommendations.

9. L249 Please explain the abbreviations glucagon-like peptide-1 (GLP-1) and others as you do in L255-256

Response: These abbreviations have been explained upon first appearance in the manuscript as suggested.

10. L299 CO2 and O2 – subscript; Energy expenditure (EE)

Response: Thank you for noticing this error. We have reformatted these for subscripts.

11. L302 EE and RQ, respectively ?: 

Response: We have added this statement in the manuscript to improve clarification as suggested.

12. L326 Name that group (CONTROL)

Response: We have named the group from which this cat was removed (CONTROL) for clarity as suggested.

13. L333 and L335 shouldn't be " <" for Ptime ?

Response: Thank you for bringing this error to our attention. We have fixed this to be the correct order.

---

## [Editor Report · Decision Letter 1]

9 Feb 2022

Dietary choline in gonadectomized kittens improved food intake and body composition but not satiety, serum lipids, or energy expenditure

PONE-D-21-36602R1

Dear Dr. Adronie Verbrugghe,

We’re pleased to inform you that your manuscript has been judged scientifically suitable for publication and will be formally accepted for publication once it meets all outstanding technical requirements.

Kind regards,

Ewa Tomaszewska, DVM Ph.D

Academic Editor

PLOS ONE